# Trends in Vitamin C Consumption in the United States: 1999–2018

**DOI:** 10.3390/nu13020420

**Published:** 2021-01-28

**Authors:** Mary Brauchla, Mark J. Dekker, Colin D. Rehm

**Affiliations:** PepsiCo, 700 Anderson Hill Road, Purchase, NY 10577, USA; Mary.Brauchla@pepsico.com (M.B.); Mark.Dekker@pepsico.com (M.J.D.)

**Keywords:** vitamin C, dietary surveys, trends, descriptive studies, United States, fruit juice, fruit, vegetables

## Abstract

Low intakes of fruits and vegetables have resulted in suboptimal intakes of several micronutrients, including vitamin C. This cross-sectional study used data from 84,902 children/adults (≥1 y) who completed a 24-h dietary recall as part of the United States National Health and Nutrition Examination Survey (1999–2018). Mean vitamin C intakes from foods/beverages were calculated as were trends in major food/beverage sources of vitamin C. Percentages below the Estimated Average Requirement (EAR) were estimated. Overall, mean vitamin C consumption declined by 23% (97–75 mg/d; *p*-value for trend < 0.001). 100% fruit juice was the leading source of vitamin C (25.6% of total or 21.7mg/d), but this declined by 48% (25–13 mg/d; *p*-value for trend < 0.001). Whole fruit increased among children/adolescents (+75.8%;10–17 mg/d; *p*-value for trend < 0.001), but not adults, while the vegetable contribution was generally unchanged. The proportion of the population below the EAR increased by 23.8% on a relative scale or 9 percentage points on an absolute scale (38.3–47.4%). Declines in vitamin C intake is driven largely by decreases in fruit juice coupled with modest increases in whole fruit. Due to associations between vitamin C intake and numerous health outcomes these trends warrant careful monitoring moving forward.

## 1. Introduction

Most of the United States (US) population currently fails to meet fruit and vegetables recommendations provided by the most recent Dietary Guidelines for Americans (DGAs) [1]. Recent data from the National Health and Nutrition Examination Survey (NHANES) suggests the average fruit and vegetable intake in US adults is 0.9 cup equivalents (cup eq.) and 1.5 cup eq., respectively, which is far below the recommended 1.5–2 cup eq. of fruits and 2–3 cup eq. of vegetables [2,3]. Low consumption of fruits and vegetables have led to low intake of several vitamins and minerals, including vitamin C [1,4]. Many fruits and vegetables contain high concentrations of vitamin C and commonly promoted sources in the US include red and green bell peppers, citrus fruit and juices, strawberries and kiwi [5]. Vitamin C is also added to foods and beverages to improve the nutrient profile or for other purposes, including flavoring and food preservation.

The health benefits of vitamin C are well documented: vitamin C is essential to normal function of the immune system [6,7,8], skin health and collagen synthesis [9], and function of the nervous system [10]. Meta-analyses of epidemiologic studies have shown that vitamin C measured via self-report and in sera is associated with a decreased risk of coronary heart disease, stroke and some cancers, as well as all-cause mortality [11] although whether these associations are directly related to vitamin C itself or driven by vitamin C as a marker of fruit and vegetable intake is less certain [12]. In order to ensure normal function of the body and to avoid vitamin C deficiency current US vitamin C recommendations are to consume 75 mg/day for women and 90 mg/day for men [13]. Severe vitamin C deficiency is rare but if vitamin C consumption is limited or nonexistent for several weeks it can result in scurvy, a disease causing anorexia, poor wound healing, and tooth loss and gingival bleeding [14].

Despite the clear benefits of adequate vitamin C intake this nutrient has consistently been classified as underconsumed in the DGAs since 2005 [1,15,16], though not as a nutrient of public health concern. Surprisingly, there is little data documenting the most commonly consumed sources of vitamin C and how changes in intake of top sources affect intakes. A better understanding of vitamin C consumption patterns and trends could provide valuable information to promote intake in the future and monitor for any upticks in potential sub-optimal intakes. The purpose of this study was to track vitamin C intake, percentage of the population consuming amounts below the Estimated Average Requirement (EAR) and to identify top food and beverage sources of vitamin C in a large nationally-representative sample of the US population over a 20-year period. 

## 2. Materials and Methods

### 2.1. Data Sources

This cross-sectional study was based on data from the 10 most recent 2-year cycles (1999–2018) of the continuous NHANES. NHANES data is a nationally representative survey that uses a complex multistage probability sample to create a representative sample of the noninstitutionalized civilian US population. NHANES is the flagship survey for assessing the diets of the US population. The response rates to the survey differ by year, but were 62.8% in 2017–2018 and 82% in 1999–2000 [17]. The National Center for Health Statistics (NCHS) obtained Institutional Review Board approval and informed consent was obtained from all subjects; the data have subsequently been made freely available for public use [18]. 

### 2.2. Dietary Recall Data

For primary analyses, dietary data came from a single 24-h dietary recall conducted in-person. A single 24-h recall for a large sample size will lead to an unbiased estimate of population-level average intakes [19]. Respondents reported the types and amounts of all foods and beverages consumed in the preceding 24 h, from midnight to midnight in an in-person dietary recall using a computerized assisted Automated Multiple Pass Method. The method probes for commonly forgotten foods and queries detailed information on the amounts of foods consumed using common reference units and examples [20]. For young children (<6 y) the parent was the primary respondent; for children 6–11 y, the child was the primary respondent, but a parent/guardian was present and could assist. For children aged ≥12 y, the child was the primary source of dietary recall, but could be assisted by an adult. Dietary recalls were conducted in either Spanish or English.

### 2.3. Analysis Approach

Secondary analyses examined the proportion of the population consuming vitamin C levels below pre-specified thresholds (e.g., Estimated Average Requirement). Because a single 24-h recall (or even the average of multiple recalls) cannot reliably estimate the population distribution of intakes we used the National Cancer Institute (NCI) method to estimate the proportion of the population consuming intakes below this threshold. As this method requires a subset of participants to have a second 24-h recall we used data from 2003-onwards for these analyses (1999–2002 public-use data do not include a second dietary recall). The second dietary recall is conducted over the telephone after the initial 24-h recall.

To examine trends in vitamin C intake stratified by food/beverage source (hereafter referred to as source) data were broken into five mutually exclusive groups: 100% fruit juice, whole fruit, vegetables, fruit drinks and other sources. These categories were created based on preliminary analyses examining the top sources of vitamin C in the diet. Each category was defined based on the prefix code within the Food and Nutrient Database for Dietary Studies [21]. For example, foods starting with a “7” were identified as vegetables (including vegetable juices), those starting as “611”, “642” and “643” were identified as 100% fruit juice, and “9252”, “9253”, “9255” among others were identified as fruit drinks. Because mixed dishes may contain many of these food groups (e.g., apple pie or mixed meat dishes would be classified as “other” but would include vitamin C from apples and vegetables respectively). As such, these categories should be considered as generally representative and not a full accounting of the total contribution of fruits, vegetables or fruit juices. It was not possible to use the What We Eat in America food categories as this database is only available from 2005 onwards.

Analyses were conducted for the total population age ≥1 y who had at least one valid 24-h dietary recall as defined by NCHS staff. Per standard analysis procedures of 24-h dietary recall data, there was no additional exclusion of individuals based on extremely low or high energy intakes. Pregnant and lactating women were included in the analysis of dietary recalls.

### 2.4. Biochemical Indicators of Vitamin C Intake

Secondary analyses compared biochemical indicators of vitamin C measured in sera for 2003–2006 (pooled from two cycles) and 2017–2018 among individuals with valid serum vitamin C levels (age ≥ 6 y) participating in the laboratory assessment of NHANES. Pregnant women were excluded. The protocols for data collection and analyses were generally comparable across the years and included both fasted and unfasted samples [22,23]. Consistent with prior analyses, vitamin C levels were transformed using a square-root transformation and back-transformed for data presentation [24,25]. Because multiple factors can influence vitamin C measurements and these factors may have changed over time (e.g., decreasing smoking and increasing obesity) both crude and adjusted analyses were conducted [26,27]. Adjusted analyses included age (using unrestricted splines), gender, race/ethnicity, family income-to-poverty ratio (with a missing indicator), measured body mass index (BMI) and an indicator of smoking status derived from serum cotinine levels. As analyses included both children and adults, a combined BMI measure was created that grouped people into an underweight (<18.5 for adults; <5% ile for children), healthy weight (18.5–24.9; 5–84.9% ile), overweight (25–29.9; 85–94.9% ile) and obese (≥30; ≥95% ile) based on standard cut-points for adults and percentiles from the Centers for Disease Control and Prevention (CDC) growth charts for children. Current smokers were defined as individuals with a serum cotinine level ≥10 ng/mL consistent with the prior literature, with all others considered non-smokers [24]. Serum cotinine data was used instead of self-report data as consistent self-report data was not available for children/adolescents and smoking status is very strongly associated with serum vitamin C levels. Use of supplements and consumption of vitamin C in the diet was not adjusted for as they are on the causal pathway in assessing any potential trends in vitamin C levels.

### 2.5. Statistical Analyses

For each 2-year cycles, the survey-weighted mean and corresponding 95% confidence interval of average vitamin C intakes from sources was calculated. A survey-weighted linear regression model was then used to determine if there was a significant linear trend in intakes of vitamin C. Analyses were repeated for age/sex groups defined by the Dietary Reference Intakes (DRI) publication [28]. Analyses by source of vitamin C were conducted in a similar manner and for the overall population, children and adolescents (age 1–18 y) and adults (≥19 y) separately. The NCI method was used to estimate the proportion of the population both overall and by age/sex population sub-groups from the DRI [29]. For analyses of serum vitamin C both crude and adjusted models were implemented using survey-weighted linear regression models of the square-root transformed serum vitamin C level, with results back-transformed for interpretation. A complete case analysis approach was used for the serum vitamin C analysis (with missing indicators for BMI and poverty status). Given the large sample size and number of statistical tests being employed a two-sided α-level of 0.01 was used to determine statistical significance. Primary analyses used Stata 16.0 (College Station, TX, USA) and implementation of the NCI method was done in SAS 9.4 (SAS Institute Inc., Cary, NC, USA) using macros developed and publicly available on the NCI website [30]. All analyses appropriately accounted for the complex survey design of NHANES data.

## 3. Results

For primary analyses, the sample size was 84,902 (range across survey cycles 7284 to 9322. Overall sample characteristics across the 20-year study period are shown in Table 1, which as per design of the survey show the data are representative of the US population. The mean vitamin C levels are shown for each socio-demographic group for reference. Young children and adolescents and younger adults had the highest average intakes of vitamin C, and males consumed about 15.2% more vitamin C (though this difference is explained by energy intakes; on an energy-adjusted basis, female participants consumed more vitamin C [87.4 mg/d vs. 74.8 mg/d per 2000 kcal]). By race/ethnicity, the non-Hispanic white population consumed the least vitamin C and the Mexican-American population consumed the most, though these differences are largely attributable to differences in the underlying age distributions of the different populations. The highest income individuals consumed marginally more vitamin C than other groups, but there was no clear gradient by income.

Trends in mean vitamin C intakes over the 20-year period are shown in Table 2. On average, vitamin C levels decreased by about 2.3 mg/d for each 2-year period or by 22.6% overall (*p*-value for trend < 0.001). Declines were particularly dramatic among children 1–3 y (change from 1999–2000 to 2017–2018: −32.1%), 14–18y females (−39.6%), 14–18y males (−33.6%), women 51–70 y (−30.7%), and men 51–70 y (−25.7%). The only sub-groups not experiencing a statistically significant decline were 9–13y males and females, 31–50 y females and older adults (≥71 y). Because other changes in the composition of the population can affect trends in dietary intakes, sensitivity analyses evaluated overall trends adjusted for age, sex, race/ethnicity and energy intakes. There was no observable difference between adjusted and unadjusted results (see Table 3). 

Trends in sources of vitamin C are provided in Figure 1. Overall (Figure 1A), the decline in vitamin C intakes appears to be driven by decreases in vitamin C from 100% fruit juice and fruit drinks. In 1999–2000, 100% fruit juice was the second leading source of vitamin C (just below fruit), but in 2017–2018, it was the fourth leading source of vitamin C, declining by nearly half (−48.4%). At the same time, vitamin C from vegetables was generally unchanged and vitamin C from whole fruit increased by about 25.8%. The contribution of fruit drinks to vitamin C intakes also declined dramatically by more than half (−50.7%). Vitamin C from other sources did not materially change. Some differences in trends by sources were observed when data were stratified by age. Among children/adolescents (Figure 1B), the contribution of whole fruit to vitamin C intakes increased, whereas no increase was observed for adults (Figure 1C). While the contribution of juice declined in both groups, the decline was more pronounced among adults. For children/adolescents there was a modest decline in vitamin C from vegetables and some suggestion of the same for adults, but not statistically significant.

### 3.1. EAR Results

For the overall population, the proportion of the population below the EAR increased from 38.3% to 47.4% from 2003–2004 to 2017–2018, a 24% relative increase (Figure 2A). Figure 2B shows the proportion of the population below the EAR by age/sex group in 2003–2004 compared to 2017–2018. The most marked increase in the proportion below the EAR was among the youngest (1–3 y), females 14–18 y, males 14–18 y, males 19–30 y, males 31–50 y and older males. The only sub-group with a decrease in the proportion below the EAR were children 4–8 y.

### 3.2. Trends Based on Biomarkers of Vitamin C

A total of 20,675 individuals age ≥ 6 y (6696 in 2017–2018 and 13,979 in 2003–2006) were included in the analysis of serum vitamin C levels. The results of the serum vitamin C analysis diet are shown in Figure 3. Overall, there was a non-significant decline in serum vitamin C levels in fully-adjusted analyses of approximately 4.8% from 2003–2006 to 2017–2018. There was little suggestion of a difference in serum vitamin C levels for children and adolescents. For adults, there was some evidence of a decrease (declined by approximately 5.5%), but like the overall analysis, this difference was not statistically significant.

## 4. Discussion

In this large nationally-representative survey of more than 84,000 participants with high-quality dietary data collected over a 20-year period we observed a 22.6% decrease in vitamin C intakes. This decrease was observed amongst all but a limited number of age/sex groups and appears to be driven by reduced contribution of vitamin C from 100% fruit juice and no marked change in vitamin C from vegetables. Vitamin C from whole fruits increased for children/adolescents, but not adults. From 2003–2004 to 2017–2018, the proportion of individuals with intakes below the EAR increased from 38.3% to 47.4%. Analyses of serum vitamin C observed a non-significant decrease in vitamin C levels, particularly for adults, but the proportional decline was less than was observed for diet.

While far from optimal, prior population-based studies have shown that the US-diet has modestly improved over the past 15–20 years. Notable changes include increased whole grain, fiber, polyunsaturated fat and whole fruit consumption and decreased consumption of added sugars and solid fats [31]. Fiber (+16.6% for adults; +25.3% for children/adolescents) and calcium (+20.1% for adults; +21.2% for children), two nutrients of public health concern, increased among adults and children from 1999–2012 and 1999–2016, respectively [31,32]. Vitamin D intakes have not meaningfully changed among adults [31] and no major trend was observed for potassium. The current finding of fairly-dramatic decreases in vitamin C intakes in the context of improving diet quality overall represents somewhat of a paradox.

One possible explanation for these findings is the reduced consumption of 100% fruit juices. Our results show that in 1999–2000, about 26% of vitamin C consumed in the diet came from 100% fruit juice, declining to 17% by 2017–2018. This finding is consistent with a large body of research demonstrating decreased consumption of 100% fruit juice [31,32,33,34,35]. To date, studies documenting changes in fruit juice consumption have not formally assessed the potential impact on nutrient intakes. While many professional organizations recommend whole fruit over 100% fruit juice, juice is a beverage and is more practically replaced by another beverage such as water, juice drinks and other beverages or none-at-all rather than a direct exchange for whole fruit [36,37]. Outside of milk (where consumption has dropped precipitously), 100% fruit juices tend to have a very high nutrient density compared to other beverages [38].

While not directly quantifiable, much of these consumption changes may be due to messaging around the sugar content of juice and purported links between sugar and adverse health outcomes [39] which have also translated to recomendations from the American Academy of Pediatrics suggesting limiting juice intake in children to certain age-specific levels [40]. The observed change in whole fruit intake is less consistent on the population level with children increasing intake far more than adults. This difference may be due to national efforts such as revamping the school lunch program and initiatives like “Let’s Move” and “Play 60” ad campaigns that are specifically targeted towards children. Curiously, trends in whole fruit consumption show increasing intakes among US adults, at least through 2011–2012. This suggests that the increase in whole fruit was not for vitamin C rich fruits (e.g., citrus, berries or tropical fruits) and may have been for fruits with relatively little vitamin C (e.g., apples or bananas), the latter of which are amongst the most commonly consumed fruits in the US [41]. This observation highlights the need to conduct detailed analyses that not only examine trends in dietary intakes by broad food sources but using as fine-resolution as possible to truly understand the dietary changes being made in the population.

While the changes in vitamin C sources from juice and whole fruit were compelling it is worth noting that we observed no marked change in vitamin C contributions from vegetables, which is consistent with relatively flat consumption of vegetables over this study period. For children, we observed a modest trend towards less vitamin C from vegetables and for adults there was some suggestion of such a trend (*p*-value for trend = 0.09). The generally flat intake of vegetables represents an important public health challenge as vegetables have numerous health benefits and are nutrient dense and energy sparse. Multiple campaigns and policies have been implemented to encourage vegetable consumption but there remain numerous barriers to intake. The high cost of whole fruit and vegetables, particularly fresh versions, is one often-cited barrier [42,43]. Descriptive studies show that on a per-calorie basis these are amongst the costliest foods [44] and qualitative studies show that consumers, especially those with lower-incomes, are highly attuned to their cost [45]. Beyond their direct cost, availability challenges are noteworthy, along with concerns regarding taste, quality, and time of preparation/convenience [46,47,48]. In part due to these barriers, individuals with higher socioeconomic status (SES) routinely consume greater amounts of whole fruits and vegetables than individuals of lower SES [45,49]. However, all groups consume too few fruits and vegetables, so simply eliminating disparities in intake will not be adequate to bring population-level intakes in-line with recommendations. On the other-hand numerous relatively inexpensive 100% fruit juices, are widely available. United States Department of Agriculture estimates show a median price per fruit cup equivalent of $0.32 for frozen juices from concentrate and $0.42 for ready-to-drink juices, compared to $0.72, $1.18, $0.94 and $0.91 for fresh, frozen, canned and dried fruits respectively [50]. In terms of food loss, the percent of consumer losses for fresh fruits and vegetables stands at 33 and 30% respectively, compared to 13% for processed fruit (a broader category which includes juice) [51]. Future studies should be conducted to unpack the specific mechanisms informing dietary choices for vitamin C dense foods and beverages.

This analysis focused on vitamin C from foods and beverages and deliberately did not include vitamin C from supplements. This choice was made because numerous professional organizations, including the American Heart Association and American Institute for Cancer Research encourage individuals to meet their micronutrient recommendations through diet alone [52,53]. The Academy of Nutrition and Dietetics and the US Preventive Services Task Force do not go quite this far but both discourage supplement use for the purposes of disease prevention [54,55] which stands in contrast to the most commonly reported reason reported for supplement use, which is to “improve/maintain health” [56]. An analysis of NHANES data from 1999–2012 among adults, showed the 30-day use of vitamin C supplements, including from multi-vitamins decreased significantly from 42% to 36% [57], but more recent data are not available. Published data on trends among children/adolescents are not available but data from 2017–2018 NHANES show only 3.0% were taking a single ingredient vitamin C supplement and 23.8% were taking some type of multivitamin/multimineral supplement, which usually contain vitamin C [58].

It is important to consider the declines in vitamin C intakes observed here in the context of dietary adequacy. The issue with declining trends in vitamin C intakes from foods/beverages is not a concern of acute deficiency (e.g., scurvy), but rather on its potential impact on the immune system, skin health and collagen synthesis, nervous system and potentially on reactive oxidative species [59]. On a comparative basis the proportion below the EAR for vitamin C is higher than some nutrients of public health concern. Cowan et al. show in sex-stratified analyses that the proportion of men with vitamin C intakes less than the EAR (50.8% in 2011–2014) is higher than calcium (26%) and potassium (35%), but much lower than vitamin D (91.5%) [60]. For women, the proportion below the EAR was higher for vitamin C than for potassium, though more women were below the EAR for calcium [60]. Another reason to monitor this decline is the connection between vitamin C intake and iron bioavailability. In the latest Dietary Guidelines for Americans, iron was identified as an under-consumed nutrient for certain groups [1] and vitamin C is known to increase the bioavailability of non-heme iron [61]. As consumer interest in a “flexitarian” or “plant-based” diet has increased along with decreasing trends of some animal foods this is an issue meriting careful assessment [62,63]. Given these concerns, the decrease in mean vitamin C intakes and more importantly, the increase in the proportion of individuals with intakes below the EAR is of potential concern and merits careful monitoring looking forward.

Study limitations include the use of self-reported dietary data which is subject to systematic and random errors [64]. Despite this, professionally collected 24-h recalls are considered amongst the strongest dietary assessment instruments and form the cornerstone of dietary surveillance in the US and elsewhere. Further, a single 24-h recall can provide an unbiased estimate of population average intakes. Our analysis of biochemical indicators of diet may be subject to residual confounding, via covariate measurement error or inefficient parameterization of variables, though we did adjust for variables previously shown to be strongly related to serum vitamin C and both BMI and serum cotinine are objective measures not subject to self-report bias, though they may be subject to other forms of measurement error. Strengths of the study included a very large sample size, which allowed us to examine statistically reliable data by two-year cycle and present data stratified by age and sex. A further strength of the study is looking at the specific sources of changes in vitamin C, which allow us to start identifying the potential reasons for trends rather than simply describing them.

## 5. Conclusions

In this large population-based study we observed a dramatic decline in vitamin C intakes that appears to be driven by declines in fruit juice consumption. A modest increase in vitamin C from whole fruit could not make up the gap due to declines in 100% fruit juice consumption. While at the population-level there is little risk of severe vitamin C deficiency, given observed relationships and associations between vitamin C intake and numerous health outcomes these trends are of potential concern. The proportion of individuals with vitamin C intakes less than the EAR appear to be greater than for other nutrients of public health concern including calcium (for men) and potassium. There is tremendous need to increase fruit and vegetable consumption in the United States and multiple approaches will likely be necessary to impart these changes. Looking forward, vitamin C intakes should be continually monitored and identifying specific sub-populations at greater risk for inadequate vitamin C intakes could be identified. Close examinations of dietary trends for specific dietary constituents of interest beyond the overall topline trend can help us understand why the diet may be changing and identify potential strategies to improve diet.

## Figures and Tables

**Figure 1 nutrients-13-00420-f001:**
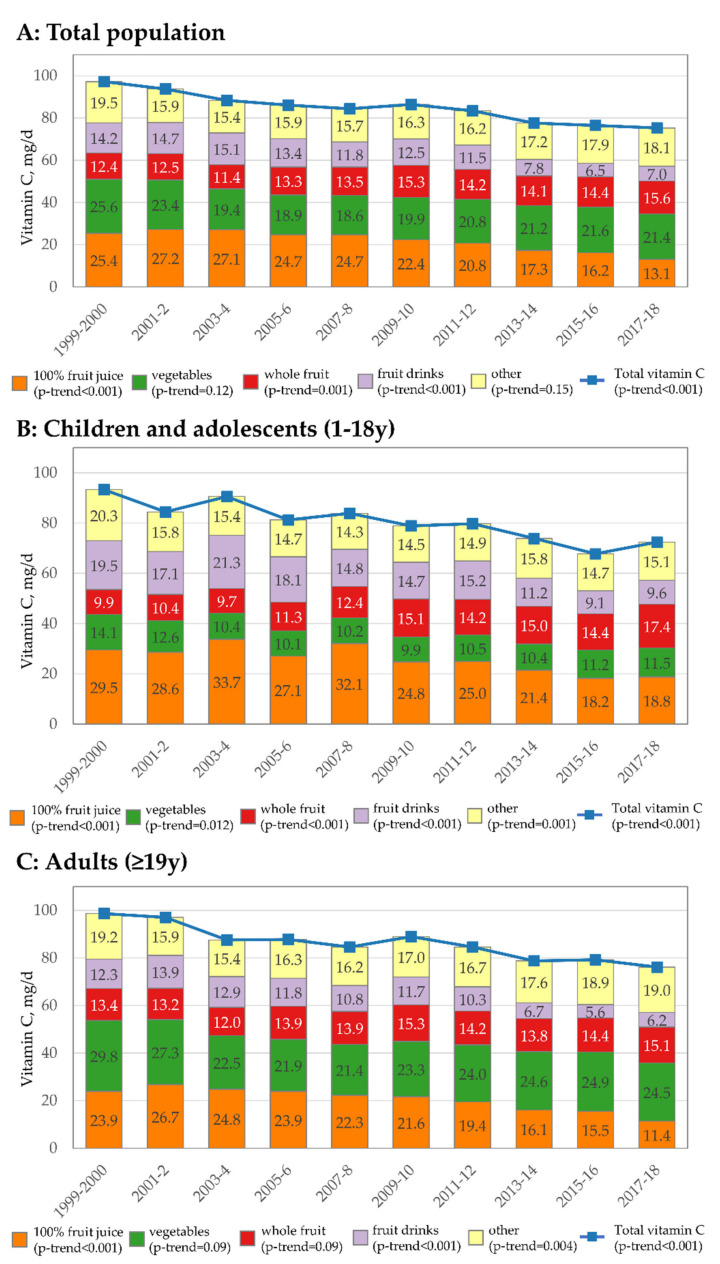
Trend in vitamin C intakes overall and by specific food/beverage source overall (**A**), among children/adolescents 1–18 y (**B**), and adults ≥ 19 y (**C**), 1999–2018.

**Figure 2 nutrients-13-00420-f002:**
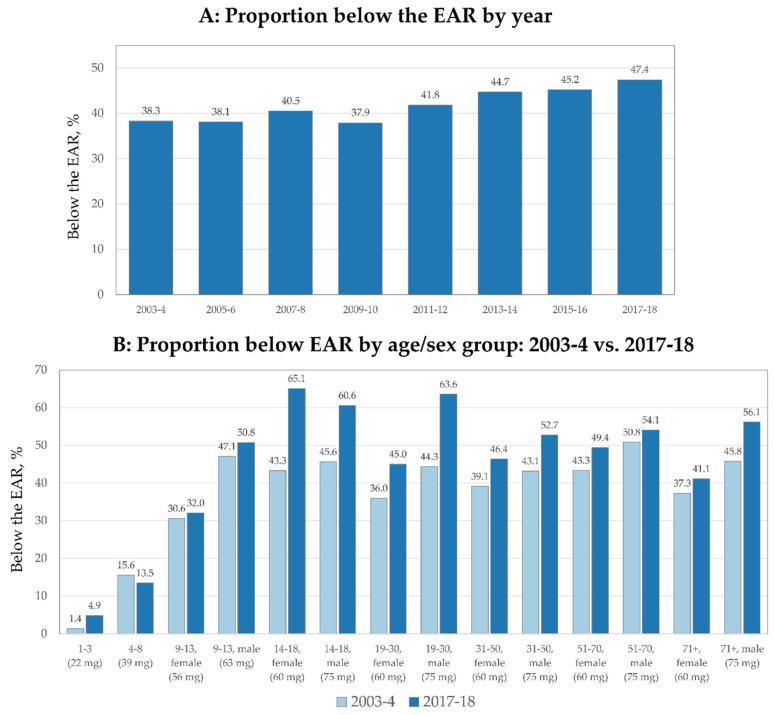
Trends in proportion of population below the Estimated Average Requirement (EAR) overall (**A**) and by age/sex group (**B**).

**Figure 3 nutrients-13-00420-f003:**
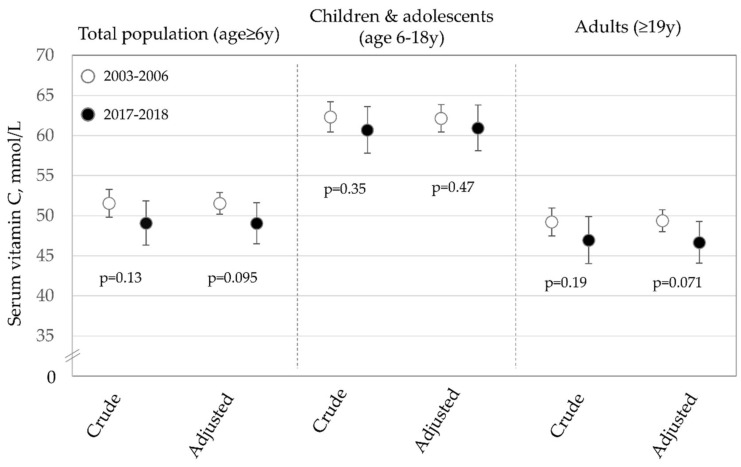
Mean serum vitamin C concentrations in 2003–2006 compared to 2017–2018 overall, among children/adolescents and adults (Adjusted model includes covariates for age group, race/ethnicity, sex, BMI, family income-to-poverty ratio and serum cotinine levels. Overall sample size is 20,675, 6821 for children/adolescents and 13,854 for adults).

**Table 1 nutrients-13-00420-t001:** Population characteristics, 1999–2018.

	*N*	Weighted %	Mean Vitamin C, mg/d (95% CI)
Total	84,902	100.0	85 (83, 86)
Age group			
1–3	6605	4.0	88 (85, 91)
4–8	8534	6.8	81 (78, 84)
9–13	9219	6.9	75 (73, 78)
14–18	9976	7.1	81 (77, 84)
19–30	11,211	17.0	87 (84, 90)
31–50	16,322	27.6	86 (83, 89)
51–70	15,214	22.4	85 (83, 88)
≥71	7821	8.1	85 (82, 88)
Sex			
Female	43,138	51.2	79 (77, 80)
Male	41,764	48.8	91 (89, 93)
Race/ethnicity			
Non-Hispanic White	32,117	65.3	80 (78, 82)
Non-Hispanic Black	20,007	12.0	94 (92, 97)
Mexican-American	18,577	9.8	95 (91, 98)
Other Hispanic	6801	5.7	92 (89, 96)
Other race/mixed race	7400	7.1	87 (83, 90)
Family income-to-poverty ratio ^a^			
<1.00 [lower income]	20,621	16.2	85 (82, 88)
1.00–1.99	20,957	20.1	80 (77, 83)
2.00–3.99	19,880	26.5	81 (79, 83)
≥4.00 [higher income]	16,625	30.2	90 (87, 93)
Missing	6819	6.9	87 (94, 91)

^a^ Interpreted as ratio of family income to the federal poverty guidelines. In 2018, the federal poverty level for a family of four in the contiguous United States was $25,100.

**Table 2 nutrients-13-00420-t002:** Trends in population mean vitamin C consumption from foods and beverages (mg/d) overall and by Dietary Reference Intake age and sex groups, 1999–2018.

		Vitamin C mg/d (95% CI)										*p*-Value for Trend
	EAR (mg)	1999–2000	2001–2002	2003–2004	2005–2006	2007–2008	2009–2010	2011–2012	2013–2014	2015–2016	2017–2018	
Total population (age ≥ 1 y)	-	97 (90, 105)	94 (87, 100)	88 (83, 94)	86 (83, 89)	84 (77, 91)	86 (84, 89)	83 (77, 90)	78 (75, 80)	77 (72, 81)	75 (71, 80)	<0.001
Age/sex groups												
1–3	22	109 (95, 123)	92 (86, 98)	105 (93, 116)	84 (75, 94)	102 (91, 112)	81 (74, 88)	85 (71, 99)	81 (75, 88)	68 (61, 76)	74 (63, 84)	<0.001
4–8	39	93 (85, 101)	80 (71, 90)	91 (83, 99)	80 (70, 91)	86 (77, 94)	79 (71, 86)	84 (76, 92)	72 (66, 79)	67 (55, 78)	79 (68, 90)	<0.001
9–13, female	56	77 (68, 87)	82 (71, 92)	84 (74, 94)	76 (66, 86)	67 (53, 81)	73 (62, 85)	69 (58, 80)	64 (57, 72)	64 (59, 68)	80 (62, 98)	0.08
9–13, male	63	81 (68, 94)	81 (69, 92)	77 (67, 87)	73 (63, 83)	87 (73, 101)	68 (55, 81)	80 (70, 90)	80 (69, 92)	73 (61, 84)	70 (56, 83)	0.21
14–18, female	60	91 (84, 99)	78 (63, 92)	77 (68, 86)	74 (64, 84)	77 (65, 89)	73 (62, 85)	59 (54, 64)	65 (54, 76)	66 (54, 79)	55 (46, 63)	<0.001
14–18, male	75	107 (89, 125)	97 (84, 110)	106 (80, 132)	98 (87, 108)	79 (69, 90)	98 (83, 114)	96 (58, 135)	79 (66, 92)	70 (60, 80)	71 (54, 88)	<0.001
19–30, female	60	87 (76, 97)	88 (77, 98)	82 (70, 94)	78 (71, 85)	80 (66, 95)	80 (72, 89)	80 (69, 92)	71 (66, 75)	75 (65, 85)	71 (62, 80)	0.002
19–30, male	75	100 (84, 116)	123 (93, 154)	94 (81, 108)	106 (90, 121)	96 (82, 111)	101 (86, 116)	96 (78, 114)	87 (75, 100)	87 (75, 99)	71 (61, 81)	<0.001
31–50, female	60	87 (72, 102)	82 (69, 96)	85 (73, 96)	76 (63, 89)	75 (63, 87)	80 (71, 88)	76 (68, 84)	74 (66, 82)	73 (64, 82)	74 (67, 82)	0.028
31–50, male	75	109 (96, 122)	107 (89, 125)	99 (84, 115)	95 (87, 103)	95 (83, 108)	97 (85, 108)	86 (73, 99)	76 (70, 82)	84 (74, 95)	85 (72, 98)	<0.001
51–70, female	60	101 (89, 112)	94 (87, 101)	78 (68, 87)	78 (71, 85)	82 (72, 91)	91 (78, 105)	76 (69, 82)	75 (69, 80)	74 (66, 82)	70 (62, 77)	<0.001
51–70, male	75	109 (96, 122)	101 (92, 110)	87 (80, 94)	98 (89, 108)	85 (75, 95)	91 (84, 98)	99 (83, 114)	83 (75, 90)	83 (73, 93)	81 (72, 90)	<0.001
≥71, female	60	94 (85, 103)	85 (75, 94)	83 (76, 89)	81 (75, 88)	74 (68, 80)	78 (72, 85)	81 (71, 92)	85 (78, 92)	76 (64, 87)	75 (63, 87)	0.031
≥71, male	75	112 (99, 125)	90 (82, 99)	87 (79, 96)	98 (84, 112)	86 (78, 95)	89 (75, 103)	83 (69, 96)	97 (85, 109)	88 (74, 101)	85 (71, 99)	0.06

EAR refers to Estimated Average Requirement.

**Table 3 nutrients-13-00420-t003:** Sensitivity analyses for overall trend adjusted for covariates.

	Vitamin C mg/d (95% CI)										*p*-Value for Trend
	1999–2000	2001–2002	2003–2004	2005–2006	2007–2008	2009–2010	2011–2012	2013–2014	2015–2016	2017–2018	
Unadjusted	97 (90, 105)	94 (87, 100)	88 (83, 94)	86 (83, 89)	84 (77, 91)	86 (84, 89)	83 (77, 90)	78 (75, 80)	77 (72, 81)	75 (71, 80)	<0.001
Age-adjusted	97 (90, 105)	94 (88, 100)	88 (83, 94)	86 (83, 90)	84 (77, 91)	86 (84, 89)	83 (77, 90)	78 (75, 80)	76 (72, 81)	75 (71, 79)	<0.001
Age/sex-adjusted	97 (90, 105)	94 (88, 100)	88 (83, 94)	86 (83, 90)	84 (77, 92)	86 (84, 89)	83 (77, 90)	77 (75, 80)	76 (72, 81)	75 (71, 80)	<0.001
Age/sex/race adjusted	98 (91, 105)	94 (89, 100)	89 (84, 94)	87 (83, 90)	85 (78, 91)	86 (84, 89)	83 (77, 89)	77 (75, 80)	76 (71, 80)	74 (70, 79)	<0.001
Energy- adjusted	94 (87, 101)	90 (84, 96)	84 (78, 89)	83 (79, 86)	83 (76, 90)	85 (82, 87)	80 (74, 87)	76 (74, 78)	76 (71, 80)	73 (69, 77)	<0.001
Fully-adjusted	94 (88, 101)	91 (85, 96)	84 (80, 89)	83 (79, 87)	83 (77, 89)	85 (82, 87)	79 (73, 86)	75 (73, 78)	74 (70, 79)	72 (67, 76)	<0.001

Age defined as DRI age groups, race/ethnicity as five groups (non-Hispanic white, non-Hispanic black, Mexican-American, other Hispanic and other/mixed race). Energy parameterized as 2000 kcal. Estimated as adjusted marginal means.

## Data Availability

All data for this project are publicly available on the National Center for Health Statistics website, available at: https://wwwn.cdc.gov/nchs/nhanes/Default.aspx.

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
