# Peer review of "Trends in Vitamin C Consumption in the United States: 1999–2018"

_nutrients, 2021, doi:10.3390/nu13020420_

Round 1

Reviewer 1 Report

An interesting and worthwhile paper.

I have no major concerns, but I do note that there is a fair bit of discussion based around juice intake, and while I do appreciate it's relevance to the paper as it appears to be a driver in the reduced Vitamin C consumption, the paragraph of lines 299-307 stood out to me. After previously mentioning that the American Academy of Pediatrics recommend limiting juice intake in children and discussion of the high energy density and messaging of the sugar content of juice (lines 264-276), this paragraph seemed somewhat bias. I do not necessarily recommend removing this paragraph, but it may be worth keeping in mind that I noted the authors are employees of PepsiCo as is appropriately stated under 'conflicts of interest', and that may have influenced how I read this section. I recommend to revisit this paragraph. 

I would like to see it stated whether or not the serum for data in figure 3 was fasting or non-fasting samples, whether or not any participants were taking vitamin C containing supplements, and if this was accounted for or not in the serum analysis. 

Author Response

I have no major concerns, but I do note that there is a fair bit of discussion based around juice intake, and while I do appreciate it's relevance to the paper as it appears to be a driver in the reduced Vitamin C consumption, the paragraph of lines 299-307 stood out to me. After previously mentioning that the American Academy of Pediatrics recommend limiting juice intake in children and discussion of the high energy density and messaging of the sugar content of juice (lines 264-276), this paragraph seemed somewhat bias. I do not necessarily recommend removing this paragraph, but it may be worth keeping in mind that I noted the authors are employees of PepsiCo as is appropriately stated under 'conflicts of interest', and that may have influenced how I read this section. I recommend to revisit this paragraph. 

We thank the reviewer for their thoughtful comments regarding this paragraph. Based on the reviewer feedback we have made numerous modifications to the language here and in a few other places.

I would like to see it stated whether or not the serum for data in figure 3 was fasting or non-fasting samples, whether or not any participants were taking vitamin C containing supplements, and if this was accounted for or not in the serum analysis. 

The data for serum vitamin C was from both fasted and non-fasted samples. Prior analyses of serum vitamin C data from NHANES (including those referenced in the paper) did not limit to fasted samples. We have added this information to the methods in the appropriate section. In the current paper, we discuss that we did not adjust for use of vitamin C supplements as this behavior would be on the causal pathway of any observed trends.

Reviewer 2 Report

First of all, I thank the editors for being able to review this manuscript.
The present manuscript entitled "Trends in vitamin C consumption in the United States: 1999-2018" is an article of great interest and relevance, despite the interest for readers, the manuscript has certain limitations that are developed below:

- There are acronyms that are not developed, such as US. As the lines are not numbered, it cannot be indicated. This acronym is at the beginning of the introduction.
- The material and methods section should be divided into subsections, numbered following the journal's guidelines.
- In the material and methods section, the type of statistical program with which the data analysis was carried out is not specified.
- In the results section, the numerous tables make it difficult to understand the study. It would be advisable to reduce them to the interest and objective of the study, or to transfer them as annexes.
- In these tables it would be advisable to add a table footer, in which the acronyms used are specified.
- In the figures, the format should be taken care of and the same font and size should be established as the rest of the manuscript.
- In the bibliographic references section, the citations do not follow the regulations of the journal.

Author Response

First of all, I thank the editors for being able to review this manuscript.
The present manuscript entitled "Trends in vitamin C consumption in the United States: 1999-2018" is an article of great interest and relevance, despite the interest for readers, the manuscript has certain limitations that are developed below:

- There are acronyms that are not developed, such as US. As the lines are not numbered, it cannot be indicated. This acronym is at the beginning of the introduction.

We thank the reviewer for bringing this to our attention. In re-submitting the manuscript we have carefully checked to ensure that all acronyms are explained at their first use.

- The material and methods section should be divided into subsections, numbered following the journal's guidelines.

We agree with the reviewer’s suggestion and have implemented this change.

- In the material and methods section, the type of statistical program with which the data analysis was carried out is not specified.

If we are understanding correctly, the reviewer’s comment refers to the specific statistical software this information is provided on page 4 of the original manuscript as copied below. If the reviewer is referring to specific commands (e.g., “svy: mean” or “svy: proportion”) this information feels excessively detailed and anybody familiar with the analysis of data from complex surveys should be able to identify the commands used. However, we have added some additional information regarding the application of the NCI method in SAS as that is a more specialized procedure.

“Primary analyses used Stata 16.0 (College Station, TX) and implementation of the NCI method was implemented in SAS using macros developed and publicly available on the NCI website [30]. All analyses appropriately accounted for the complex survey design of NHANES data.”

- In the results section, the numerous tables make it difficult to understand the study. It would be advisable to reduce them to the interest and objective of the study, or to transfer them as annexes.

We thank the reviewer for this comment. In our experience, readers often skip supplemental information and the number of tables/figures is not more than what one normally observes in an original investigation. The only table that would be appropriate to move to a supplement is Table 3 and it does not feel appropriate to have a 1-table appendix.

- In these tables it would be advisable to add a table footer, in which the acronyms used are specified.

We have implemented this suggestion.

- In the figures, the format should be taken care of and the same font and size should be established as the rest of the manuscript.

We have implemented this suggestion within the space constraints of the format. In some cases the font is slightly smaller than the main text but we have increased it as much as is possible.

- In the bibliographic references section, the citations do not follow the regulations of the journal.

We thank the reviewer for pointing this out and have modified the citations accordingly.

Reviewer 3 Report

This work is distinguished by long-term research and provides valuable insight into trends in vitamin C intake.

The Reviewer indicated a few remarks.

  • Line 36-38: "Vitamin C is also occasionally added to foods and beverages to improve the nutrient profile or for other purposes, including flavoring and food preservation." - Vitamin C is a commonly used antioxidant as a food additive.
  • Why is Table 2 duplicated?
  • Please put chart captions below them. Please increase the font size of Figure 1 and 2.The Reviewer proposes to add data labels for all bars as different changes over the years have been examined (Figure 1).
  • "The current finding of fairly-dramatic decreases in vitamin C intakes in the context of improving diet quality overall represents somewhat of a paradox."- Please describe possible causes, influencing factors, make hypotheses. In addition to changes in the quality of life and mentions of consumer education, please refer to the processing factors in the fruit and vegetable industry, the quality of products, and progress in production technology.
  • Please indicate the direction of future research on the subject of this paper.

Author Response

This work is distinguished by long-term research and provides valuable insight into trends in vitamin C intake.

The Reviewer indicated a few remarks.

  • Line 36-38: "Vitamin C is also occasionally added to foods and beverages to improve the nutrient profile or for other purposes, including flavoring and food preservation." - Vitamin C is a commonly used antioxidant as a food additive.

We have deleted the term “occasionally” and agree with the reviewer comment. Thank you for bringing this to our attention.

  • Why is Table 2 duplicated?

The duplicated table 2 was labeled in error and has been re-labeled correctly as “Table 3”, we thank the reviewer for pointing this out.

  • Please put chart captions below them. Please increase the font size of Figure 1 and 2.The Reviewer proposes to add data labels for all bars as different changes over the years have been examined (Figure 1).

Within the constrains of the space provided we have made the suggested edits to the graphs.

  • "The current finding of fairly-dramatic decreases in vitamin C intakes in the context of improving diet quality overall represents somewhat of a paradox."- Please describe possible causes, influencing factors, make hypotheses. In addition to changes in the quality of life and mentions of consumer education, please refer to the processing factors in the fruit and vegetable industry, the quality of products, and progress in production technology.

Thank you for your comment.  We have clarified in the revised manuscript to clarify that one of our explanations for the apparent paradox is reduced consumption of 100% fruit juice without subsequent compensation by other sources of vitamin C.  Although elements of processing, quality and production could certainly influence micronutrient content on a small scale, we feel the broader shift in consumption patterns is primarily responsible for these results.

  • Please indicate the direction of future research on the subject of this paper.

We have added this information in numerous spots in the discussion. Specifically, when talking about the EAR results we have added:

“Given these concerns, the decrease in mean vitamin C intakes and more importantly, the increase in the proportion of individuals with intakes below the EAR is of potential concern, merits careful monitoring looking forward and should be the focus of further study.”

When discussing potential barriers we have added the following text, “Future studies should be conducted to unpack the specific mechanisms informing dietary choices for vitamin C rich foods.”